# Antimicrobial resistance and prevalence of extended-spectrum beta-lactamase-producing *Klebsiella* species in East Tennessee dairy farms

Benti Deresa Gelalcha,[1] Aga E. Gelgie,[1] Oudessa Kerro Dego[1]

**ABSTRACT**  *Klebsiella* species commonly reside in dairy cattle guts and are consistently exposed to beta-lactam antibiotics, including ceftiofur, which are frequently used on the U.S. dairy farms. This may impose selection pressure and result in the emergence of extended-spectrum beta-lactamase (ESBL)-producing strains. However, information on the status and antimicrobial resistance (AMR) profile of ESBL-*Klebsiella* spp. in the U.S. dairy farms is largely unknown. This study aimed to determine the prevalence and AMR profile of ESBL-*Klebsiella* spp. and the factors affecting their occurrence in dairy cattle farms. Rectal fecal samples ($n = 508$) and manure, feed, and water samples ($n = 64$) were collected from 14 dairy farms in Tennessee. Samples were directly plated on CHROMagar ESBL, and presumptive *Klebsiella* spp. were confirmed using matrix-assisted laser desorption/ionization-time of flight mass spectrometry. Antimicrobial susceptibility testing was performed on the isolates against panels of 14 antimicrobial agents from 10 classes using minimum inhibitory concentration. Of 572 samples, 57 (10%) were positive for ESBL-*Klebsiella* spp. The fecal prevalence of ESBL-*Klebsiella* spp. was 7.2% (95% CI: 6.5–8.0). The herd-level fecal prevalence of ESBL-*Klebsiella* spp. was 35.7% (95% CI: 12.7–64.8). The fecal prevalence of ESBL-*Klebsiella* spp. was significantly higher in calves than in cows and higher in cows with higher parity ($\geq3$) as compared to cows with low parity ($P < 0.001$). Most (96.5%, $n = 57$) ESBL-*Klebsiella* spp. were resistant to ceftriaxone. The highest level of acquired co-resistance to ceftriaxone in ESBL-*Klebsiella* spp. was to sulfisoxazole (66.7%; 38/57). About 19% of ESBL-K*lebsiella* spp. were multidrug resistant. The presence of ESBL-producing *Klebsiella* spp. in dairy cattle, feed, and water obtained from troughs could play a crucial epidemiological role in maintaining and spreading the bacteria on farms and serving as a point source of transmission.

**IMPORTANCE**  We collected 572 samples from dairy farms, including rectal feces, manure, feed, and water. We isolated and identified extended-spectrum beta-lactamase (ESBL)-*Klebsiella* spp. and conducted an antimicrobial susceptibility test and analyzed different variables that may be associated with ESBL-*Klebsiella* spp. in dairy farms. The results of our study shed light on how ESBL-*Klebsiella* spp. are maintained through fecal-oral routes in dairy farms and possibly exit from the farm into the environment. We determine the prevalence of ESBL-*Klebsiella* spp. and their antimicrobial susceptibility profiles, underscoring their potential as a vehicle for multiple resistance gene dissemination within dairy farm settings. We also collected data on variables affecting their occurrence and spread in dairy farms. These findings have significant implications in determining sources of community-acquired ESBL-*Enterobacteriaceae* infections and designing appropriate control measures to prevent their spread from food animal production systems to humans, animals, and environments.

Address correspondence to Oudessa Kerro Dego, okerrode@utk.edu.

The authors declare no conflict of interest.

**KEYWORDS**   antimicrobial resistance, dairy farm, extended-spectrum beta-lactamase, *Klebsiella* species, prevalence

Antimicrobial resistance (AMR) is recognized as one of the top five global public health threats of this century (1). The overuse and/or misuse of antimicrobial drugs in multiple sectors, including humans, animals, and agriculture, is a key driver for the rapid increase of AMR (2–6). In the United States, beta-lactam antibiotics, especially third-generation cephalosporins (3GCs), are frequently used for the treatment and prevention of diseases in dairy cattle farms (7–11) and in human health settings (12, 13). Both veterinary and human 3GCs have similar chemical structure, mode, and spectrum of action and are thus susceptible to extended-spectrum beta-lactamase (ESBL) enzymes (14, 15) that hydrolyze the β-lactam ring of the 3GCs (16). ESBL-producing Gram-negative bacteria especialy *Enterobacteriaceae* are mostly multidrug resistant (MDR) (17–22). The use of the same generation of cephalosporins in dairy cattle farms and human health services may lead to cross-resistance (10, 23).

Previously, most ESBL-*Enterobacteriaceae* infections were associated with human health care (24, 25). Recent reports showed a change in the epidemiology of these infections, where nearly half of the cases were community associated (26). The U.S. Centers for Disease Control and Prevention (CDC) also reported a 9% average annual increase of hospitalized patients from community-acquired ESBL pathogens in 6 consecutive years (2012–2017) (26). The CDC also estimated 197,400 cases of ESBL-*Enterobacteriaceae* among hospitalized patients and 9,100 estimated deaths in the U.S. alone (26). The primary sources of community-associated ESBL-producing *Enterobacteriaceae* are unknown. Among *Enterobacteriaceae*, *Escherichia coli* and *Klebsiella* species are the most frequently identified bacteria carrying ESBL-encoding genes (16, 27–30). Food animal farms, especially dairy farms, may serve as reservoirs of ESBL-*Klebsiella* species because of the frequent use of 3GCs (25, 26, 31–33). This study provides evidence that *Klebsiella* species isolates from dairy cattle and associated farm samples carry high burden of multidrug-resistance genes including ESBL genes. Thus, dairy farms may serve as one of the sources of community-acquired ESBL-*Klebsiella* species infections in humans through direct contact with carrier animals and/or indirect transmission through food chain or environmental sources.

*Klebsiella* species belongs to the genus *Klebsiella* in the family of *Enterobacteriaceae*. *Klebsiella* is an opportunistic pathogen and a natural inhabitant of the gastrointestinal tract (GIT) of healthy animals and humans (34). *Klebsiella* spp. are a common member of dairy cattle's normal gut microbiota, and a carriage rate as high as 100% has been reported in the U.S. (35). In addition, *Klebsiella* spp. are frequently isolated from dairy cattle's normal gut and other body parts, including teat end (36, 37), indicating contamination of the udder and teat with bacteria shed through feces into the environment. In addition to being a common colonizer of animal's GIT, some *Klebsiella* spp. are among the primary causes of coliform mastitis in dairy cattle (38). *Klebsiella pneumoniae* and *Klebsiella oxytoca* are among the most frequent causes of environmental mastitis and mastitis-related culling (35–38).

As a common inhabitant of dairy cattle's GIT, *Klebsiella* spp. are often exposed to beta-lactam antibiotics, especially to 3GCs, which are used for the treatment and prevention of mastitis (8, 39). This may lead to the development of resistance to beta-lactam antibiotics, including ceftiofur (38), mainly through the production of ESBL enzymes (40).

A recent study on *Klebsiella* isolates from cow with mastitis in Wisconsin reported a significant resistance to ceftiofur, a common antibiotic used to treat and prevent cases of mastitis (41). However, the authors reported no significant changes in resistance trends against ceftiofur over the years. A similar recent study on *Klebsiella* spp. isolates from milk of cows with clinical mastitis and milk with high somatic cell counts found high resistance against cephalothin and ceftiofur (38). Another recent whole genome

sequencing-based study on milk from cases of mastitis in dairy farms from Iowa State reported the isolation of ESBL-*K. pneumoniae* (42).

Despite the widespread presence of *Klebsiella* spp. in dairy cattle and the extensive use of ceftiofur in dairy farms that may exert selection pressure,these bacteria have received none to very little attention regarding their status as an ESBL producer. For example, until recently, *Klebsiella* spp. were not included in the National Antibiotic Resistance Monitoring System (NARMS). As a result, information on the burden and factors affecting the occurrence of ESBL-*Klebsiella* spp. on the U.S. dairy farms is unknown. Therefore, the objective of this study was to determine the prevalence, antimicrobial resistance, and factors affecting the occurrence of ESBL-*Klebsiella* spp. in conventional dairy farms.

## MATERIALS AND METHODS

### Study design and sample collection

A cross-sectional study was conducted on 14 conventional (non-organic) dairy farms in East Tennessee, selected for their proximity to the University of Tennessee and the high concentration of dairy farms in this region. The farms were randomly chosen from a pool of 108 dairy farms in the area. Initially, 60 farms were selected, and attempts were made to contact each of them to gauge their interest in participating. Ultimately, from 50 farms that we were able to contact, only 14 of them agreed to participate in the study.

Individual animal rectal fecal samples (dairy cows, $n = 424$; calves, $n = 84$) and farm environmental samples (manure, $n = 30$; feed, $n = 15$ and water, $n = 19$) were collected from 14 dairy farms in East Tennessee. We collected about 100 and 20 g of individual rectal feces from dairy cows and calves, respectively, into 50-mL sterile falcon tubes (Thermo Fisher Scientific, Waltham, MA, USA). We also collected 200 g of manure and feed each and 50 mL of water from troughs, into separate 50-mL sterile Falcon tubes (Thermo Fisher Scientific). All samples were immediately placed on ice in a cooler box and transported to the lab and processed within 24 h of collection.

We also collected a brief survey on general farm management and antibiotic usage. In addition, we recorded specific details about each sampled animal, including age, parity, breed, physiological status (dry or lactating), recent treatment history, and any use of ceftiofur within the past 6 months prior to sampling. This study was approved by the University of Tennessee's Institutional Animal Care and Use Committee (IACUC) Registration Number 2782-0720.

### Isolation and identification of ESBL-*Klebsiella* species

In the lab, each 10 g of individual animal feces, feed, and manure samples was mixed manually by gloved hand with 90 mL of Tryptic Soy Broth (TSB-PO4) (MG Scientific, Pleasant Prairie, WI, USA) in a Whirl-Pak bag (Whirl-Pak, Pleasant Prairie, WI, USA) for 15–20 seconds. Similarly, 20 mL of water sample was mixed with 80 mL of TSB-PO4. The mixture was kept at room temperature for 2 h. We plated 50 µL of the mixture on CHROMagar ESBL plates (DRG International Inc., Springfield, NJ, USA) and incubated at 37°C for 24 h to isolate ESBL-*Klebsiella* species. Presumptive ESBL-*Klebsiella* sp. colonies (metallic blue) were subcultured onto new CHROMagar ESBL plates (DRG International) to get a pure and well-isolated colony. Presumptive ESBL-*Klebsiella* spp. were confirmed using matrix-assisted laser desorption/ionization-time of flight Mass Spectrometry (MALDI-TOF MS) as described by the manufacturer (Bruker Daltonics, Billerica, MA, USA).

### Antimicrobial susceptibility testing

*Pseudomonas aeruginosa* ATCC 27853, *Escherichia coli* ATCC 25922, *Staphylococcus aureus* ATCC 29213, and *Streptococcus pneumoniae* ATCC 49619 were used as quality control strains. All MALDI-TOF MS confirmed isolates were tested for antimicrobial susceptibility

againstpanels of 14 antimicrobial agents using the broth microdilution method. The 14 antimicrobials include β-lactams (ampicillin, ceftriaxone, cefoxitin, amoxicillin-clavulanic acid, and meropenem), aminoglycosides (gentamicin and streptomycin), folate pathway inhibitors (sulfisoxazole, trimethoprim/sulfamethoxazole), macrolides (azithromycin), phenicols (chloramphenicol), quinolones (ciprofloxacin and nalidixic acid), and tetracyclines. The minimum inhibitory concentration (MIC) of the antimicrobials was determined using Sensititre plates (Sensititre CMV4AGNF) (Thermo Fisher Scientific, CA, USA) following the manufacturer's protocol and the Clinical and Laboratory Standards Institute guidelines (CLSI M100: *Enterobacteriaceae*) (43).

## Statistical data analyses

We collected raw data and imported it into Microsoft Excel for Windows 10 (Microsoft Corp., Redmond, WA) before transferring it to SPSS (SPSS Statistics for Windows, Version 27.0, IBM Corp, Armonk, NY) for statistical analysis. The unit of analysis was the sample, dairy cattle, farm, and bacterial isolate. The prevalence of ESBL-*Klebsiella* spp. from different sources (units) was determined. Farms were classified as ESBL-*Klebsiella* spp. positive if the bacteria were detected on CHROMagar ESBL from the fecal sample of at least one dairy cattle. Descriptive (frequency) statistics were used to summarize the collected data, such as proportions across various categories. Inferential (mixed-effects logistic regression) statistics were used to analyze the data and evaluate the association between selected predictors and the odds of ESBL-*Klebsiella* sp. fecal carriage. The significance level for all statistical analyses was set at 0.05.

## RESULTS

### Microbiological results

Based on the results of the phenotypic screening of samples on the chromogenic agar (CHROMagar ESBL), 221 metallic blue colonies were recovered from rectal fecal, manure, water, and feed samples. Out of 221 isolates, 25.8% (*n* = 57 isolates) was confirmed to be *Klebsiella* spp. by MALDI-TOF MS. The rest were *Citrobacter sedlakii* (*n* = 132 isolates), *Citrobacter amalonaticus* (*n* = 7), *Citrobacter koseri* (*n* = 5), *Citrobacter freundii* (*n* = 3), *Enterobacter cloacae* (*n* = 8), *Enterobacter bugandensis* (*n* = 3), *Enterobacter asburiae* (*n* = 2), and *Proteus vulgaris* (*n* = 4). From 57 *Klebsiella* spp., the majority, 87.7% (50/57) of isolates, were obtained from rectal feces, whereas the remaining 12.3% (7 isolates) comprised of two isolates each from feed and water and three isolates from pooled manure samples. We identified four different *Klebsiella* spp. using MALDI-TOF MS. These include *K. pneumoniae* (91%; *n* = 52), *Klebsiella variicola* (*n* = 2), *K. oxytoca* (*n* = 2), and *Klebsiella aerogenes* (*n* = 1).

### Fecal prevalence of ESBL-*Klebsiella* species at farm level

The weighted fecal prevalence of ESBL-*Klebsiella* spp. in dairy cattle was 7.2% (95% CI: 6.5–8.0). *Klebsiella* spp. were recovered from at least one fecal sample of cattle in five farms (*n* = 14), resulting in herd-level fecal prevalence of 35.7% (95% CI: 12.7–64.8). The within-herd prevalence of fecal ESBL-*Klebsiella* among positive herds ranged from 4% to 62.4%. Eighty percent (*n* = 57) of fecal and 73.6% of all *Klebsiella* sp. isolates were obtained from a single farm, Farm M (Table 1).

ESBL-*Klebsiella* spp. were recovered from all sample types (Table 2). Furthermore, the prevalence of ESBL-*Klebsiella* spp. in combined samples of manure, water, and feed (10.9%, *n* = 64) was not significantly different (*P* = 0.22) from that of rectal fecal samples (9.8%), as determined by mixed-effects logistic regression analysis (result not shown).

### Factors affecting fecal ESBL-*Klebsiella* species prevalence in dairy cattle

Parity was a risk factor for fecal carriage of ESBL-*Klebsiella* spp. The odds of cows with parity ≥ 3 were about five times [OR$_{adj}$ = 4.9 (95% CI: 3.0–8.03), *P* < 0.001] of cows with

**TABLE 1** Within-herd prevalence of fecal ESBL-*Klebsiella* species[b]

| County[a] | Farm | Herd size | Sample size (n) | Unweighted frequency and prevalence: n (%) | 95% CI | Weighted frequency and prevalence (%): n/N (%) | 95% CI |
|---|---|---|---|---|---|---|---|
| A | A | 100 | 14 | 2 (14.3) | 1.8–42.8 | 14 (14) | 7.9–22.4 |
| A | B | 14 | 9 | 3 (33.3) | 7.5–70 | 5 (35.7) | 12.7–64.9 |
| B | C | 750 | 63 | 0 (0) | NA | 0 (0) | NA |
| B | D | 300 | 30 | 0 (0) | NA | 0 (0) | NA |
| C | E | 50 | 13 | 0 (0) | NA | 0 (0) | NA |
| C | F | 110 | 40 | 0 (0) | NA | 0 (0) | NA |
| D | G | 170 | 25 | 0 (0) | NA | 0 (0) | NA |
| E | H | 35 | 26 | 0(0) | NA | 0 (0) | NA |
| F | I | 61 | 30 | 0 (0) | NA | 0 (0) | NA |
| G | J | 60 | 25 | 3 (12) | 2.5–31.2 | 7 (11.7) | 4.8–22.6 |
| G | K | 1,700 | 45 | 0 (0) | NA | 0 (0) | NA |
| F | L | 250 | 50 | 2 (4) | 0.49–13.7 | 10 (4) | 1.9–7.2 |
| H | M | 450 | 64 | 40 (62.5) | 49.5–74.3 | 281 (62.4) | 57.8–66.9 |
| F | N | 350 | 74 | 0 (0) | NA | 0 (0) | NA |
| Total | | 4,400 | 508 | 50 (9.8) | 7.3–12.8 | 318 (7.2) | 6.5–8.0 |

[a]Farms from counties designated by the same letter are in the same county.
[b]NA, not applicable.

parity $\leq$ 2 to carry ESBL-*Klebsiella* spp. in their feces. The age and parity of cows were strongly correlated ($R = 0.73$; $P < 0.001$); thus, the age of the cows was not included in the analysis.

The weighted fecal prevalence of ESBL-*Klebsiella* spp. was 13.3% (95% CI: 11.1–15.7) in calves and 6.2% (95% CI: 5.4–7.1) in adult cows. The odds of recovering ESBL-*Klebsiella* spp. from calf feces were about 10 times higher than those of cow feces [$OR_{crude} = 10.4$ (5.7–18.8), $P < 0.001$] (Table 3). Among dairy cattle, ESBL-*Klebsiella* spp. were isolated from only lactating cows and none from dry cows. Similarly, all ESBL-*Klebsiella* isolates were obtained from animals with no recent treatment history with 3GCs. Thus, further analysis was not necessary on these two variables.

## Antimicrobial resistance profile of ESBL-*Klebsiella* species

As expected, all 57 (100%) of ESBL-*Klebsiella* sp. isolates were resistant to ampicillin. All but two isolates (96.5%; 55/57) from five sample sources were resistant to ceftriaxone. Out of the 14 antimicrobial agents evaluated in the study, the two isolates exhibited resistance only to ampicillin and sulfisoxazole. Both ceftriaxone-susceptible isolates were obtained from the same farm, Farm J. The highest level of acquired co-resistance among ceftriaxone-resistant *Klebsiella* spp. was seen with sulfisoxazole (66.7%; 38/57), followed by tetracycline (15.8%; 9/57) and amoxicillin-clavulanic acid (10.5%; 6/57). Only one isolate, retrieved from a cow fecal sample, exhibited co-resistance to cefoxitin. *Klebsiella* isolates recovered from calves showed the highest level of concurrent resistance against most of the tested antibiotics (Table 4).

Among the critically important antimicrobials (CIAs), ceftriaxone-resistant *Klebsiella* spp. displayed the highest levels of concurrent resistance to streptomycin (10.5%) and

**TABLE 2** Prevalence of ESBL-*Klebsiella* species in dairy cattle farms according to sample sources

| Sample source | (No. tested)[a] | (No. of positives)[b] | Prevalence (95% CI) |
|---|---|---|---|
| Rectal feces | 508 | 50 | 9.8 (7.4–12.8) |
| Manure | 30 | 3 | 10% (2–26.5) |
| Water | 19 | 2 | 10.5% (1.1–33.1) |
| Feed | 15 | 2 | 13.3% (1.7–40.5) |
| Total | 572 | 57 | 10% (7.6–12.7) |

[a]Number of samples tested for ESBL-*Klebsiella* species from each sample source.
[b]Number of ESBL-*Klebsiella* isolates obtained from each sample sources.

**TABLE 3** Relationship between ESBL-*Klebsiella* species and animal-level variables

| Variable | Category | Weighted frequency | Prevalence (95% CI) | OR (95% CI) | P value |
|---|---|---|---|---|---|
| Parity | ≤2 | 114 | 5.4% (4.4–6.3) | Ref. | |
| | >2 | 88 | 6.3% (5.1–7.7) | 4.91 (3.0–8.03) | <0.001 |
| Herd structure | Cow | 202 | 6.2% (5.4–7.1) | Ref. | |
| | Calf | 116 | 13.3% (11.1–15.7) | 10.43 (5.7–18.8) | <0.001 |
| Total | | 318 | 7.2% (6.5–8.0) | NA | |

gentamycin (8.8%). Among sample sources, ceftriaxone-resistant *Klebsiella* spp. isolated from calves showed the highest level of concurrent resistance to the highest priority CIAs: streptomycin (26.3%), azithromycin (15.8%), and gentamycin (10.5%). Acquired concurrent resistance was relatively low for ceftriaxone and fluoroquinolones (3.5%). Importantly, all *Klebsiella* sp. isolates were susceptible to meropenem.

## Multidrug-resistant *Klebsiella* spp. prevalence in dairy farms

The overall prevalence of MDR (acquired resistance to ≥3 classes of antibiotics) *Klebsiella* sp. isolates was 19.3% (11/57). The prevalence of MDR *Klebsiella* spp. recovered from calves (31.6%; 6/19) was higher than that retrieved from cows (12.9%; 4/31) though the difference was not statistically significant after controlling for the farm effect ($P > 0.05$) (data not shown). The prevalence of MDR *Klebsiella* isolates resistant to at least five classes of antibiotics was 36.4% (4/11), and all of them were recovered from rectal fecal samples collected from the same farm (Farm M). Three-fourths of *Klebsiella* spp. isolated from calves were resistant to at least five classes of antibiotics. Most MRD (90%;10/11) *Klebsiella* spp. were isolated from rectal fecal samples, and most of them (80%) were from the same farm (Farm M). No MDR *Klebsiella* spp. were detected in the water and feed samples (Fig. 1).

## Distribution of antimicrobial-resistant ESBL-*Klebsiella* spp. across farms

*Klebsiella* spp. from Farm M were resistant to 11 out of 14 antimicrobial agents tested in this study, and some resistance patterns were limited to this farm. MDR ESBL-*Klebsiella*

**TABLE 4** Frequency of resistant ESBL-*Klebsiella* species ($n = 57$) to 14 antimicrobial agents[c]

| Antibiotic | Sources of *Klebsiella* isolates and proportion of resistance to the specific antibacterial agent | | | | | Overall |
|---|---|---|---|---|---|---|
| | Cows ($N = 31$)[a] n (% Rest.)[b] | Calf ($N = 19$) n (% Rest.) | Manure ($N = 3$) n (% Rest.) | Feed ($N = 2$) n (% Rest.) | Water ($N = 2$) n (% Rest.) | n (% Rest.) |
| AUG2 | 2 (6.5) | 2 (10.5) | 1 (33.3) | 1 (50) | 0 | 6 (10.5) |
| AMP | 31 (100) | 19 (100) | 3 (100) | 2 (100) | 2 (100) | 57 (100) |
| AZM | 0 | 3 (15.8) | 0 | 0 | 0 | 3 (5.3) |
| FOX | 1 (3.2) | 0 | 0 | 0 | 0 | 1 (1.8) |
| AXO | 31 (100) | 19 (100) | 2 (66.7) | 2 (100) | 1 (50) | 55 (96.5) |
| CHL | 0 | 3 (15.8) | 0 | 0 | 0 | 3 (5.3) |
| CIP | 1 (3.2) | 1 (5.3) | 0 | 0 | 0 | 2 (3.5) |
| MERO | 0 | 0 | 0 | 0 | 0 | 0 (0) |
| GEN | 2 (6.5) | 2 (10.5) | 1 (33.3) | 0 | 0 | 5 (8.8) |
| NAL | 0 | 1 (5.3) | 0 | 0 | 0 | 1 (1.8) |
| STR | 1 (3.2) | 5 (26.3) | 0 | 0 | 0 | 6 (10.5) |
| FIS | 21 (67.7) | 14 (73.7) | 2 (66.7) | 1 (50) | 2 (100) | 40 (70.2) |
| TET | 3 (9.7) | 5 (26.3) | 1 (33.3) | 0 | 0 | 9 (15.8) |
| STX | 1 (3.2) | 3 (15.8) | 0 | 0 | 0 | 4 (7) |

[a]The total number of *Klebsiella* isolates obtained from a given source.
[b]Number and proportion of isolates resistant to a given antibiotic from a specific source.
[c]Rest., resistant; AXO, ceftriaxone; FOX, cefoxitin; AUG2, amoxicillin-clavulanic acid; AMP, ampicillin; MERO, meropenem; AZM, azithromycin; CIP, ciprofloxacin; NAL, nalidixic acid; CHL, chloramphenicol; GEN, gentamicin; STR, streptomycin; TET, tetracycline; FIS, sulfisoxazole; STX, trimethoprim-sulfamethoxazole.

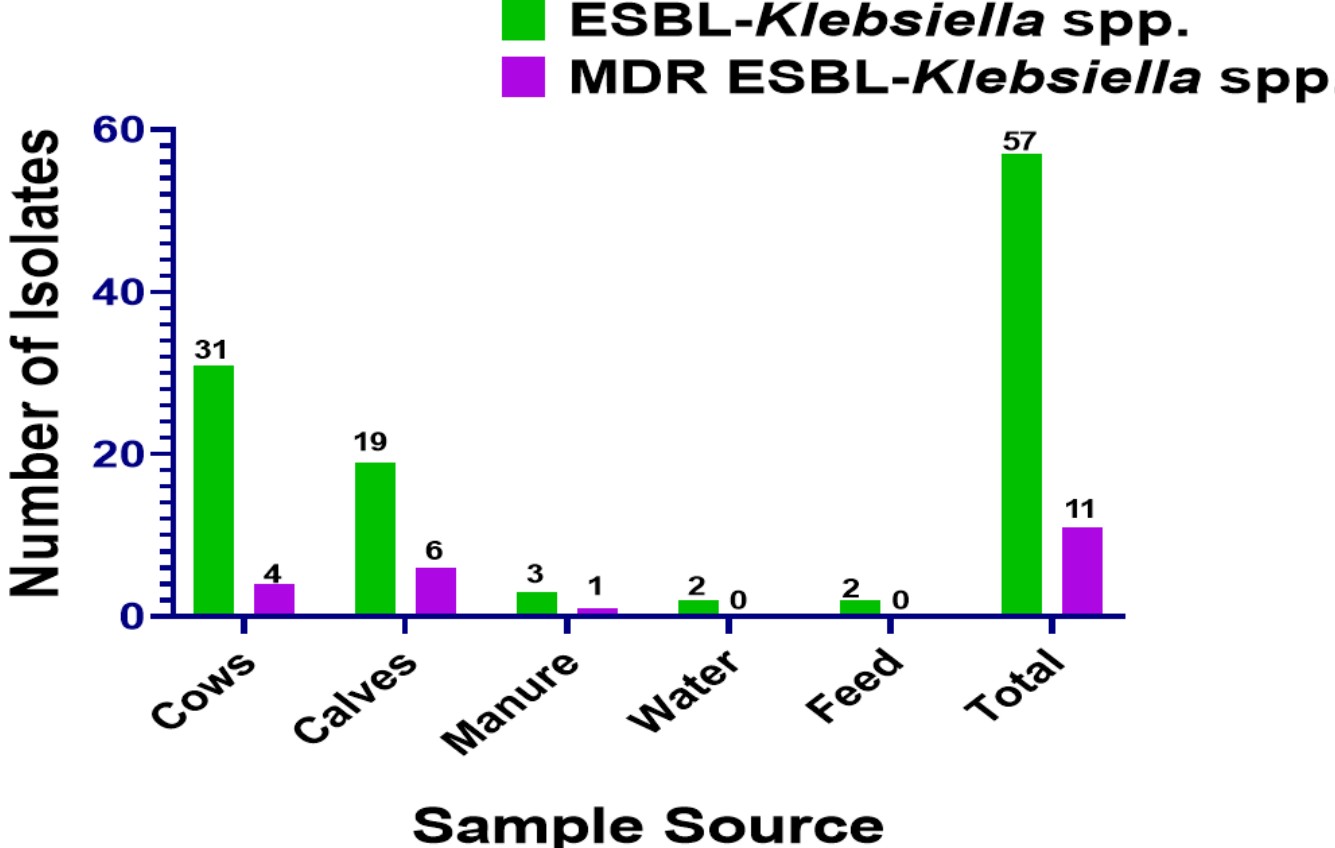

**FIG 1** Distribution of multidrug-resistant ESBL-*Klebsiella* species across the five sample sources.

spp. with concurrent resistance to six classes of antibiotics (ceftriaxone, azithromycin, cefoxitin, chloramphenicol, trimethoprim-sulfamethoxazole, and gentamicin) were obtained from farm M alone. Ceftriaxone-resistant *Klebsiella* spp. with co-resistance to sulfisoxazole were widespread and detected in all six farms (Table 5).

**TABLE 5** Distribution of antimicrobial-resistant ESBL-*Klebsiella* species ($n = 57$) across six farms

| Antibiotic | Number of isolates ($n$) resistant to a specific antibiotic in each farm | | | | | | Total |
|---|---|---|---|---|---|---|---|
| | Farm A ($n = 3$)[a] | Farm B ($n = 3$) | Farm G ($n = 2$) | Farm J ($n = 5$) | Farm L ($n = 2$) | Farm M ($n = 42$) | |
| AUG2 | 2 | 0 | 1 | 0 | 0 | 3 | 6 |
| AMP | 3 | 3 | 2 | 5 | 2 | 42 | 57 |
| AZM | 0 | 0 | 0 | 0 | 0 | 3 | 3 |
| FOX | 0 | 0 | 0 | 0 | 0 | 1 | 1 |
| AXO | 3 | 3 | 2 | 3 | 2 | 42 | 55 |
| CHL | 0 | 0 | 0 | 0 | 0 | 3 | 3 |
| CIP | 0 | 0 | 0 | 0 | 1 | 1 | 2 |
| MERO | 0 | 0 | 0 | 0 | 0 | 0 | 0 |
| GEN | 0 | 0 | 0 | 0 | 0 | 5 | 5 |
| NAL | 0 | 0 | 0 | 0 | 1 | 0 | 1 |
| STR | 0 | 0 | 0 | 0 | 1 | 5 | 6 |
| FIS | 2 | 3 | 2 | 3 | 2 | 28 | 40 |
| TET | 0 | 0 | 0 | 1 | 0 | 8 | 9 |
| STX | 0 | 0 | 0 | 0 | 0 | 4 | 4 |

[a]AXO, ceftriaxone; FOX, cefoxitin; AUG2, amoxicillin-clavulanic acid; AMP, ampicillin; MERO, meropenem; AZM, azithromycin; CIP, ciprofloxacin; NAL, nalidixic acid; CHL, chloramphenicol; GEN, gentamicin; STR, streptomycin; TET, tetracycline; FIS, sulfisoxazole; STX, trimethoprim-sulfamethoxazole.

## DISCUSSION

This study identified ESBL-producing *Klebsiella* spp. in 10% (57/572) of the collected samples and from all sample types (rectal fecal samples, pooled manure, animal feed, and water from watering troughs). The presence of ESBL-*Klebsiella* spp. in animal feed and watering troughs could play a crucial epidemiological role in maintaining and spreading the bacteria on farms and serving as a critical point source of transmission (44). Detection of ESBL-*Klebsiella* in all samples is in agreement with the previous studies that reported the pervasive presence of the bacteria in dairy cattle environments (35–37, 44).

The herd-level prevalence of ESBL-*Klebsiella* spp. was 35.7%. This result is higher than that of a study from Japan that reported a lower herd-level prevalence of 15% (3/20) (45). In this study, the within-farm fecal prevalence of the ESBL-*Klebsiella* spp. ranged from 4% to 62.4%, highlighting the variability in the ESBL-*Klebsiella* spp. prevalence across farms. This variation may be related to the difference in farm management practices, hygienic farm measures in keeping clean and dry, and antibiotic use patterns that may affect the emergence and spread of ESBL-*Klebsiella* spp. A previous study conducted in New York found a high within-herd fecal prevalence of 8%–100% of *Klebsiella* spp. in most of the studied dairy farms (46). However, it is less plausible to directly compare the current study and the New York study since the latter did not investigate the ESBL status of the bacteria.

The prevalence of ESBL-*Klebsiella* spp. in fecal samples, adjusted for sampling weight, was 7.2% (95% CI: 6.5%–8.0%). Previous studies have reported as high as 84% prevalence of fecal *Klebsiella* spp. in the U.S. and 27% in China from healthy adult dairy cattle (35, 46, 47). However, there is a lack of published research on the prevalence of ESBL-*Klebsiella* spp. in the rectal feces of dairy cattle. The fecal prevalence of ESBL-*Klebsiella* spp. in the present study is similar to a previous report (10%; 3/28) from dairy farms in Japan (45).

While the fecal prevalence of ESBL-*Klebsiella* spp. may seem lower than that of ESBL-*E.coli* reported in the U.S.(48) and other countries (45, 49, 50), however, the occurrence of ESBL-*Klebsiella* in the tested samples is concerning because of their potential to cause hard-to-treat infections in humans (26) and coliform mastitis in dairy cattle (41). It also poses a significant risk of spreading to humans, animals, and the environment. This suggests that dairy cattle may serve as potential sources of ESBL-*Klebsiella* spp.

The study showed that the odds of carrying ESBL-*Klebsiella* spp. in their feces for cows with parity ≥ 3 were about five times higher than those with parity ≤ 2. The reason for this association could be related to the fact that cows with higher parity may have more lifetime exposure to beta-lactam antibiotics for treatment of mastitis and/or other diseases of dairy cattle and more prolonged exposure to the ESBL-*Klebsiella* contaminated farm environment and other animals, which in turn increases the risk of acquiring ESBL-producing bacteria. The results of this study align with that of earlier studies that indicated older cows have a higher risk of having mastitis, which is potentially linked to increased beta-lactam antibiotic use and selection pressure that may lead to the occurrence of ESBL-producing *Enterobacteriaceae* (51–53). This finding has important implications for controlling the spread of ESBL-*Klebsiella* spp., at the farm level. Farmers and veterinarians should be aware of the increased risk of fecal carriage of ESBL-*Klebsiella* spp. in cows with higher parity and take appropriate measures to prevent their spread to humans, animals, and environments.

The present study found that the odds of recovering ESBL-*Klebsiella* spp. from calf feces were approximately 10 times higher than those of cow feces, with a crude odds ratio of 10.4 (95% CI: 5.7–18.8). We did not find published studies comparing ESBL-*Klebsiella* spp. prevalence between dairy calves and adult dairy cattle. However, this finding is consistent with previous studies that reported a higher proportion of ESBL-producing *E. coli* in calves than in older cattle (50, 54–59). The reason for this higher prevalence remains speculative, but selective pressure from feeding medicated milk replacers and/or feeding waste milk from antibiotic-treated cows or it could be

a potential fitness advantage in calves due to exposure to antibiotics early in life that might be responsible for this difference.

Among 57 ESBL-*Klebsiella* spp. isolates from fecal and other samples from farms, 55 (96.5%) were resistant to ceftriaxone. This result was predictable since isolates were initially screened on ESBL-selective media that select isolates resistant to third-generation cephalosporins. Resistance to ampicillin was observed in all ESBL-*Klebsiella* spp., as expected since *Klebsiella* spp. are intrinsically resistant to penicillin (60–62).

After ceftriaxone, ESBL-*Klebsiella* spp. showed the highest acquired co-resistance to sulfisoxazole, followed by tetracycline. This resistance pattern was widespread across six farms. This could be related to the increased usage of these antibiotics as both antibiotic classes are frequently used to treat dairy calves' digestive and respiratory diseases in the (11). This finding is consistent with previous studies from the U.S. (63, 64) and China (47) that reported a high frequency of co-resistance to sulfonamides and tetracycline in *Klebsiella* spp. A lower co-resistance frequency to sulfonamides in ESBL-*Klebsiella* spp. was reported from Brazil (9.5%) (65) and Canada (13%) (66), which could be related to the differences in the use of these antibiotics.

Detection of six (10.5%) ESBL-*Klebsiella* spp. resistant to amoxicillin-clavulanic acid is not expected as ESBL producers should be (by definition) susceptible to beta-lactamase inhibitors (clavulanic acid) (67–69). However, this finding suggests that these isolates might carry additional resistance genes mediating resistance to beta-lactamase inhibitors such as AmpC beta-lactamases, efflux pumps, or other beta-lactamase variants not inhibited by clavulanic acid (70). In addition, two *K. pneumoniae* isolates were susceptible to ceftriaxone despite being presumptively identified as ESBL producers and expected to be resistant to ceftriaxone. This may be due to the limited specificity of the CHROMagar ESBL used to screen them (71). A significant proportion (19.3%) of the ESBL-*Klebsiella* isolates were MDR-ESBL-producing strains. This finding aligns with previous results that reported ESBL phenotypes are often associated with MDR traits (10, 30, 59, 72–76). Concurrent resistance to at least six classes of antibiotics was observed in 36.4% of MDR-*Klebsiella* isolates, and 75% of them was recovered from calves. This may be related to the vulnerability of calves' microbiota to disruption by antibiotic use early in life, creating a selective pressure and conducive environment for the emergence and spread of MDR bacteria (77–79). However, additional research is needed to fully understand the underlying causes of the observed pattern.

Some MDR patterns observed in some isolates were limited to a single farm, while others were detected in multiple farms. For example, ESBL-*Klebsiella* spp. with concurrent resistance to ceftriaxone, azithromycin, cefoxitin, chloramphenicol, trimethoprim-sulfamethoxazole, and gentamicin were obtained from Farm M alone. This suggests that distinct factors could contribute to the emergence and spread of MDR bacteria in various farms. It could be due to the differences in antibiotic use and/or differences in farm management practices. For instance, using certain antibiotics in one farm but not in another could contribute to developing unique MDR patterns in the former. Similarly, varying levels of biosecurity measures (animal density or access to outside environments), use of waste milk from antibiotic-treated cows, disposal method, and use of medicated milk replacer for calves may affect the MDR patterns in ESBL-*Klebsiella* spp. (80, 81).

We found that ceftriaxone-resistant *Klebsiella* isolates from calves exhibited higher levels of concurrent resistance of 26.3%, 15.8%, and 10.5% to streptomycin, azithromycin, and gentamycin, respectively, than isolates from cows. This may be because of the higher exposure of calves to these antibiotics, leading to selective pressure and the development of resistance to these CIAs. For example, streptomycin and azithromycin are often used to treat digestive and respiratory diseases in dairy calves (11, 82). Compared with adult dairy cows which usually receive intramammary infusion for treatment and prevention of mastitis, dairy calves usually receive antibiotic treatment orally for GIT-related infections (e.g., diarrhea) and parenterally for other health issues

(e.g., pneumonia), which has a higher chance of creating selection pressure on microbes in the GIT than intramammary infusion.

We found that all ESBL-*Klebsiella* isolates were susceptible to meropenem, a relatively new beta-lactam antibiotic in the carbapenem class of antibiotics, which is considered the last-line option for treating infections caused by MDR Gram-negative bacteria, including ESBL producers (26). This result is expected since carbapenem antibiotics are not licensed for use in dairy cattle or other food animals in the U.S. (83). Unlike the present study, carbapenem-resistant *Enterobacteriaceae* were previously isolated from the feces of dairy cattle in Texas and New Mexico (84) and swine operations in the Midwest Region (85, 86). Thus, it is important to note that antibiotic susceptibility or resistance can vary by farm location, the species of farm animal population studied, and other specific factors involving each farm suggesting that the results of this study may not necessarily be equally applicable to other settings or populations.

## Conclusions

ESBL-*Klebsiella* spp. were identified in 10% of the tested samples. The prevalence of fecal ESBL-*Klebsiella* spp. was low ( 7.2%) compared to prevalence of ESBL-E.coli reported in previous studies. The fecal prevalence of ESBL-*Klebsiella* spp. was significantly higher in the calves than in the cows and higher in cows with higher parity. ESBL-*Klebsiella* spp. displayed high phenotypic co-resistance to ceftriaxone and sulfisoxazole, with almost 20% of the isolates being MDR. This study found ESBL-producing *Klebsiella* species in rectal fecal samples, pooled manure, animal feed, and watering troughs in the study dairy farms. This finding indicates that these bacteria can spread in the environment, posing potential risks to both human and animal health. Moreover, the presence of these bacteria in animal feed and watering troughs may have a significant role in their persistence and transmission on farms, serving as a potential source of infection.

## ACKNOWLEDGMENTS

The authors thank East Tennessee dairy farmers for volunteering to participate in this study.

B.D.G. was responsible for conceiving the research idea, designing the study, collecting samples, conducting bacterial isolation and identification, antimicrobial susceptibility testing, and writing and editing of the manuscript. A.E.G. contributed to sample collection and writing and editing of the manuscript. O.K.D. was responsible for the conception and design of the study, manuscript writing and editing, and overall project activity supervision.

## AUTHOR AFFILIATION

[1]Department of Animal Science, The University of Tennessee, Knoxville, Tennessee, USA

## AUTHOR ORCIDs

Benti Deresa Gelalcha  http://orcid.org/0000-0003-4845-715X
Aga E. Gelgie  http://orcid.org/0000-0003-1610-3381
Oudessa Kerro Dego  http://orcid.org/0000-0003-2108-8905

## AUTHOR CONTRIBUTIONS

Benti Deresa Gelalcha, Conceptualization, Formal analysis, Investigation, Writing – original draft, Writing – review and editing | Aga E. Gelgie, Investigation, Writing – review and editing | Oudessa Kerro Dego, Conceptualization, Project administration, Supervision, Writing – review and editing

## DATA AVAILABILITY

All data are included in the paper.

## ADDITIONAL FILES

The following material is available online.

## Open Peer Review

**PEER REVIEW HISTORY (review-history.pdf).** An accounting of the reviewer comments and feedback.

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
