## [Reviewer comments · Microbiology Spectrum]

Microbiology Spectrum

Antimicrobial Resistance and Prevalence of Extended-Spectrum Beta-Lactamase-Producing-Klebsiella Species in East Tennessee Dairy Cattle Farms

Benti Gelalcha, Aga Gelgie, and Oudessa Kerro Dego

Corresponding Author(s): Oudessa Kerro Dego, The University of Tennessee Knoxville

Review Timeline:

Submission Date:	October 2, 2023
Editorial Decision:	January 7, 2024
Revision Received:	March 13, 2024
Accepted:	July 15, 2024

Editor: John Osei Sekyere

Reviewer(s): The reviewers have opted to remain anonymous.

Transaction Report:

DOI: <https://doi.org/10.1128/spectrum.03537-23>

Re: Spectrum03537-23 (Antimicrobial Resistance and Prevalence of Extended-Spectrum Beta-Lactamase-Producing-Klebsiella Species in East Tennessee Dairy Cattle Farms)

Dear Dr. Oudessa Kerro Deogo:

Thank you for the privilege of reviewing your work. Below you will find my comments, instructions from the Spectrum editorial office, and the reviewer comments.

The reviewers have raised very serious and important concerns regarding the content, study design, and analyses of the article. I was inclined to reject the article but I would want to give you another opportunity to rectify it. Please do not return it until you can address all the concerns raised or the article will be rejected.

Revision Guidelines

Sincerely,
John Osei Sekyere
Editor
Microbiology Spectrum

Reviewer #1 (Comments for the Author):

The authors conducted a cross-sectional study of 14 dairy farms in East Tennessee to explore the prevalence, resistance characteristics, and factors affecting the occurrence of ESBL-Klebsiella strains. In short, epidemiological research is very

important and meaningful work. But I still think there are a few points that can be discussed further.

1. Why is the ESBL carrier rate significantly higher in calves than in adult cows? I think we can explore the possible causes further.
2. Why is ESBL significantly more prevalent on some farms than on others? Is farm management the main reason for this difference?
3. The authors used mixed-mode sampling to enhance understanding of the prevalence of resistance. However, rectal stool samples account for the majority of samples collected, which is potentially biased, and conclusions from other sources such as water and feed seem to lack some accuracy.
4. The mixed-effects logarithmic regression analysis method used in the study may have limitations because it assumes that the data have a normal distribution, which may not always be true in practice.
5. The study did not consider in detail other factors that may have influenced the prevalence of ESBL-Klebsiella spp., such as dairy farm management practices, hygiene practices, and antibiotic use patterns.
6. How prevalent is ESBL-Klebsiella spp. in other regions? And drug resistance patterns?

Reviewer #2 (Comments for the Author):

Summary and Conclusions: Antimicrobial Resistance and Prevalence of Extended-Spectrum Beta-Lactamase-Producing-Klebsiella Species in East Tennessee Dairy Cattle Farms. Gelalcha et al. conducted the prevalence and antimicrobial resistance of ESBL-Klebsiella species and factors affecting their occurrence in dairy cattle farms. Overall, the amount of work on this article is not very rich because that the whole research only has three main parts of bacterial isolation, identification and antimicrobial susceptibility testing. According to the title of this paper, the author hopes to elaborate the resistance and prevalence of ESBL-Klebsiella species to antimicrobials, but the investigation of drug-resistant genes and the characteristic information of drug-resistant strains, such as sequence typing, are lacking. Furthermore, the sampling area is not extensive, which also makes the article lack of representativeness.

Abstract:

Line15-40: The Abstract should be more concise, and it looks more like Content than an Abstract.

Summary and Conclusions: Antimicrobial Resistance and Prevalence of Extended-Spectrum Beta-Lactamase-Producing-Klebsiella Species in East Tennessee Dairy Cattle Farms. Gelalcha et al. conducted the prevalence and antimicrobial resistance of ESBL-Klebsiella species and factors affecting their occurrence in dairy cattle farms. Overall, the amount of work on this article is not very rich because that the whole research only has three main parts of bacterial isolation, identification and antimicrobial susceptibility testing. According to the title of this paper, the author hopes to elaborate the resistance and prevalence of ESBL-Klebsiella species to antimicrobials, but the investigation of drug-resistant genes and the characteristic information of drug-resistant strains, such as sequence typing, are lacking. Furthermore, the sampling area is not extensive, which also makes the article lack of representativeness. Therefore, it doesn't appeal to readers and lacks novelty.

Abstract:

Line15-40: The Abstract should be more concise, and it looks more like Content than an Abstract.

Reviewer #1 (Comments for the Author)

Reviewer (Rev): The authors conducted a cross-sectional study of 14 dairy farms in East Tennessee to explore the prevalence, resistance characteristics, and factors affecting the occurrence of ESBL-Klebsiella strains. In short, epidemiological research is very important and meaningful work. But I still think there are a few points that can be discussed further.

Rev: 1. Why is the ESBL carrier rate significantly higher in calves than in adult cows? I think we can explore the possible causes further.

Authors (Au): The disparity in ESBL-producing *Klebsiella* species prevalence between dairy calves and cows presents an intriguing area for further study. While hypotheses suggest factors like medicated milk replacers or milk from antibiotic-treated cows, such as colostrum, may contribute to selective pressure in calves, recent findings challenge this assumption. Studies have revealed a high prevalence of antimicrobial-resistant bacteria such as *E. coli* in young calves not exposed to such milk replacers or milk from antibiotic-treated cows, underscoring **the complexity of the issue**.

A controlled study is required to determine the factors driving the elevated prevalence of ESBL-producing *Klebsiella* species in dairy calves compared to adult cattle. This study underscores the potential significance of calves as a primary source or reservoir of ESBL-producing *Klebsiella* species, positioning them as **crucial sentinel animals** for surveillance purposes and also for targeted interventions and informed management strategies in dairy farming practices.

We kindly invite the authors to refer to these studies:

1. Berge AC, Atwill ER, Sicho WM. Animal and farm influences on the dynamics of antibiotic resistance in fecal *Escherichia coli* in young dairy calves. *Prev Vet Med.* 2005;69(1-2):25-38.
2. Springer HR, Denagamage TN, Fenton GD, Haley BJ, Van Kessel JAS, Hovingh EP. Antimicrobial Resistance in Fecal *Escherichia coli* and *Salmonella enterica* from Dairy Calves: A Systematic Review. *Foodborne Pathog Dis.* 2019;16(1):23-34.
3. Duse A, Waller KP, Emanuelson U, Unnerstad HE, Persson Y, Bengtsson B. Risk factors for antimicrobial resistance in fecal *Escherichia coli* from pre-weaned dairy calves. *J Dairy Sci.* 2015;98(1):500-16.
4. Schmid A HS, Messelhäuser U, Käsbohrer A, Sauter-Louis C, Mansfeld R. . Prevalence of extended-spectrum β -lactamase-producing *Escherichia coli* on Bavarian dairy and beef cattle farms. *Appl Environ Microbiol.* 2013;79(9):3027-

4. Awosile B, McClure J, Sanchez J, Rodriguez-Lecompte JC, Keefe G, Heider LC. Salmonella enterica and extended-spectrum cephalosporin-resistant Escherichia coli recovered from Holstein dairy calves from 8 farms in New Brunswick, Canada. J Dairy Sci. 2018;101(4):3271-84.

Rev: 2. Why is ESBL significantly more prevalent on some farms than on others? Is farm management the main reason for this difference?

Au: Yes, the observed variability in the within-farm fecal prevalence of ESBL-*Klebsiella* spp. can be potentially influenced by differences in farm management practices or molecular mechanisms, such as the presence of epidemic resistance plasmids in isolates from farms that yield the largest number of ESBL-*Klebsiella* isolates. All farm management-related factors we assessed in this study (e.g., herd size, farm type: closed vs. open, predominant breed, dry cow therapy methods: blanket vs. selective, types of antibiotics used in each farm, the presence of a separate treatment pen, and the management approach for waste milk) were not significant predictors of fecal prevalence of ESBL-*Klebsiella* species. However, it's essential to recognize that although we examined the types of antibiotics commonly used on each farm, we did not quantify the specific amounts of beta-lactams used due to the limited availability of antibiotic usage records in many farms. Additionally, while we did not conduct a formal assessment of farm hygiene, notable lapses were not observed, particularly in farms with the highest ESBL-*Klebsiella* prevalence.

Rev: 3. The authors used mixed-mode sampling to enhance understanding of the prevalence of resistance. However, rectal stool samples account for the majority of samples collected, which is potentially biased, and conclusions from other sources such as water and feed seem to lack some accuracy.

Au: The mixed-effects logistic regression analysis was exclusively applied to the weighted prevalence derived from rectal samples. Descriptive statistics were utilized to present findings from other sources, such as manure, feed, and water. It's important to acknowledge that rectal stool samples were prioritized due to their direct association with gut microbiota and antimicrobial resistance. While this approach may introduce some bias, it allows for a more targeted examination of resistance prevalence in a key reservoir. Additionally, leveraging descriptive statistics for other sample sources provides complementary insights into potential routes of antimicrobial resistance dissemination within the farm environment.

Rev: 4. The mixed-effects logarithmic regression analysis method used in the study may have limitations because it assumes that the data have a normal distribution, which may not always be true in practice.

Au: we appreciate the reviewer concern regarding the assumptions of the mixed-effects logarithmic regression analysis method employed in our study. While it's true that this method assumes a normal distribution of the data, it's important to note that mixed-effects models are robust and flexible, capable of accommodating deviations from normality through their inherent structure. Additionally, we used the analysis on weighted sample size and weighted prevalence, which increases the robustness of the statistical inference, even in the presence of non-normality.

Rev: 5. The study did not consider in detail other factors that may have influenced the prevalence of ESBL-*Klebsiella* spp., such as **dairy farm management practices, hygiene practices, and antibiotic use patterns.**

Au: We collected both farm-level and individual animal-level information. The farm level encompasses various factors such as herd size, farm type (closed vs. open), predominant breed, dry cow therapy methods (blanket vs. selective), types of antibiotics used in each farm, the presence of a separate treatment pen, and the management approach for waste milk. Despite thorough analysis, none of these farm-level variables demonstrated a significant association with the detection of ESBL-producing *Klebsiella* species in fecal samples, as evidenced by both univariate and multivariable analyses ($P > 0.05$). Consequently, they have been omitted from this report. Similarly, individual animal-related factors, including recent (up to six months prior to sampling collection) use of beta-lactam antibiotics like ceftiofur, parity, lactation status (lactating vs. dry or non-lactating), age, and breed, were examined for potential association with fecal ESBL-producing *Klebsiella* carriage. Only variables with significant associations (parity and cows vs calves) were discussed in the manuscript. We kindly invite the reviewer to look at section 3.3 of the manuscript.

Rev: 6. How prevalent is ESBL-*Klebsiella* spp. in other regions? And drug resistance patterns?

Au: while there is very limited published research on the prevalence and AMR patterns of ESBL-*Klebsiella* spp. in dairy cattle feces globally, especially in the U.S., existing studies predominantly focus on other sources, such as **mastitis milk and bulk tank milk**. For instance, a study conducted in New York reported a high (8 -100%) fecal prevalence of *Klebsiella* spp., albeit **without specific focus on ESBL-producing strains**. Similarly, another study from the U.S. and China reported 84% and 27%, respectively, of *Klebsiella* spp. prevalence in healthy adult dairy cattle, yet without **ESBL investigation**. In contrast, a study from Japan specifically examined ESBL-*Klebsiella* spp., revealing a prevalence of 10% in a relatively small sample size ($n=3/28$). This scarcity of data emphasizes the need for more comprehensive studies and the

significance of our study in contributing to the existing knowledge base. We have briefly discussed these points on page 9.

Reviewer #2 (Comments for the Author):

Reviewer (Rev): Summary and Conclusions: Antimicrobial Resistance and Prevalence of Extended-Spectrum Beta-Lactamase-Producing-Klebsiella Species in East Tennessee Dairy Cattle Farms. Gelalcha et al. conducted the prevalence and antimicrobial resistance of ESBL-Klebsiella species and factors affecting their occurrence in dairy cattle farms. Overall, the amount of work on this article is not very rich because that the whole research only has three main parts of bacterial isolation, identification, and antimicrobial susceptibility testing. According to the title of this paper, the author hopes to elaborate on the resistance and prevalence of ESBL-Klebsiella species to antimicrobials, but the investigation of drug-resistant genes and the characteristic information of drug-resistant strains, such as sequence typing, are lacking. Furthermore, the sampling area is not extensive, which also makes the article lack of representativeness. Therefore, it doesn't appeal to readers and lacks novelty. Abstract:

Authors (Au): We appreciate the reviewer's valuable insights into our manuscript. While we acknowledge the suggestion for a more comprehensive exploration of drug-resistant genes and strain characteristics, we want to emphasize the foundational focus of our study on ESBL-producing *Klebsiella* species. These fundamental aspects are crucial for understanding the prevalence and resistance dynamics of ESBL-producing *Klebsiella* species within dairy farming.

Previous research has predominantly concentrated on *clinical samples*, particularly mastitis milk, leaving the status of ESBL-producing *Klebsiella* species in apparently healthy dairy cattle largely unexplored. Our study stands out as one of the pioneering investigations into the prevalence of ESBL-*Klebsiella* species in apparently healthy cows from US dairy farms (in this aspect, the previous study focused on *E. coli* alone). Our findings revealed dairy cattle as significant reservoirs of ESBL-producing *Klebsiella* species, shedding light on the potential role of seemingly healthy dairy cows in disseminating antimicrobial resistance.

Therefore, we believe that our findings hold significant interest for readers and can serve as a foundational baseline for initiating further research into the molecular epidemiology of ESBL-producing *Klebsiella* species in dairy cattle farming and their potential impact on public health. Moving forward, we are dedicated to delving deeper into these research areas and expanding our sampling efforts to ensure the comprehensiveness and representativeness of our findings.

Abstract:

Rev: Line 15-40: The Abstract should be more concise, and it looks more like Content than an

Abstract.

Au: we summarized and shortened the abstract.

Klebsiella species commonly reside in dairy cattle guts and are consistently exposed to beta-lactam antibiotics, including ceftiofur, which are frequently used on the U.S. dairy farms. This may impose selection pressure and result in the emergence of extended-spectrum beta-lactamase (ESBL)-producing strains. However, information on the status and antimicrobial resistance (AMR) profile of ESBL-*Klebsiella* spp. in the U.S. dairy farms is largely unknown. This study aimed to determine the prevalence and AMR of ESBL-*Klebsiella* species and the factors affecting their occurrence in dairy cattle farms. Rectal fecal samples (n=508) and manure, feed, and water samples (n=64) were collected from 14 dairy farms in Tennessee. Samples were directly plated on CHROMagar™ ESBL, and presumptive *Klebsiella* spp. were confirmed using matrix-assisted laser desorption/ionization-time of flight mass spectrometry (MALDI-TOF MS). Antimicrobial susceptibility testing (AST) was performed on the isolates against panels of 14 antimicrobial agents from 10 classes using minimum inhibitory concentration. Of 572 samples, 57 (10%) were positive for ESBL-*Klebsiella* spp. The fecal prevalence of ESBL-*Klebsiella* spp. was 7.2 % (95% CI: 6.5 - 8.0). Herd-level fecal prevalence of ESBL-*Klebsiella* spp. was 35.7% (95% CI: 12.7 - 64.8). The fecal prevalence of ESBL-*Klebsiella* spp. was significantly higher in calves than in cows and higher in cows with higher parity (≥ 3) compared to those with low parity ($P < 0.001$). Most (96.5%, n=57) ESBL-*Klebsiella* spp. were resistant to ceftriaxone. The highest level of acquired co-resistance to ceftriaxone in ESBL-*Klebsiella* spp. was to sulfisoxazole (66.7%; 38/57). About 19% of ESBL-*Klebsiella* spp. were multidrug-resistant (MDR). The presence of ESBL-producing *Klebsiella* species in dairy cattle, feed, and water obtained from troughs could play a crucial epidemiological role in maintaining and spreading the bacteria on farms and serving as a point source of transmission.

Keywords: *Klebsiella* species, Extended-spectrum beta-lactamase, prevalence, dairy farm

Re: Spectrum03537-23R1 (Antimicrobial Resistance and Prevalence of Extended-Spectrum Beta-Lactamase-Producing-Klebsiella Species in East Tennessee Dairy Cattle Farms)

Dear Dr. Oudessa Kerro Dego:

I am pleased to inform you that your manuscript has been editorially accepted for publication. However, there are a few additional questions/corrections below that need to be answered before the final decision. Once these are completed, please return your submission so that I can move your paper forward to acceptance.

Please make these corrections:

1. Replace "minimum inhibitory concentration" in line 28 of the Abstract with broth microdilution
2. Delete enzymes in line 52 to avoid redundancy
3. In line 139, mixing samples by hand is not recommendable. Authors could have used a spatula or any sterile handler. This must be commented on or otherwise explained if not noted as a limitation
4. Line 252: Correct MRD to MDR

Sincerely,
John Osei Sekyere
Editor
Microbiology Spectrum

Reviewer #1 (Comments for the Author):

The author's work deserves recognition. I have always believed that epidemiological investigation is a worthwhile task.

The author's work deserves recognition. I have always believed that epidemiological investigation is a worthwhile task. This has far-reaching implications to public health and safety. I think this manuscript is acceptable.